# Perceptions of injury risk in the home and workplace in Nepal: a qualitative study

Elisha Joshi [1], Santosh Bhatta,[2] Toity Deave,[2] Julie Mytton,[2] Dhruba Adhikari,[3] Sunil Raja Manandhar,[3] Sunil Kumar Joshi[1]

EJ and SB contributed equally.

EJ and SB are joint first authors.

¹Nepal Injury Research Centre, Kathmandu Medical College Public Limited, Kathmandu, Nepal
²Faculty of Health and Applied Sciences, University of the West of England, Bristol, UK
³Mother and Infant Research Activities, Makwanpur, Bagmati, Nepal

**Correspondence to**
Ms Elisha Joshi;
ejoshi03@gmail.com

## ABSTRACT

**Objective** Injuries are a global health problem. To develop context-specific injury prevention interventions, one needs to understand population perceptions of home and workplace injuries. This study explored a range of views and perceptions about injuries in a variety of settings and identified barriers and facilitators to injury prevention.

**Design** Qualitative study: interviews and focus groups.

**Setting** Three administrative areas: Hetauda submetropolitan city, Thaha municipality and Bakaiya rural municipality in Makwanpur, Nepal.

**Participants** Nine focus groups (74 participants) and nine one-to-one interviews were completed; workers from diverse occupations, residents (slum, traditional or modern homes) and local government decision-makers participated in the study between May and August 2019. The interviews and discussions were audio-recorded, transcribed verbatim, translated to English and analysed thematically.

**Results** Six themes were developed: unsafe home and workplace environment; inadequate supervision and monitoring; perceptions that injuries are inevitable; safety takes low priority: financial and behavioural considerations; safety education and training; and government-led safety programmes and enforcement. Key barriers to injury prevention were perceived to be lack of knowledge about injury risk and preventive measures both at the community level and at the workplace. Facilitators were community-level educational programmes and health and safety training to employees and employers. Participants stressed the importance of the role of the government in planning future injury prevention programmes in different environments.

**Conclusions** This study highlighted that both home and workplace injuries are complex and multifactorial. Lack of knowledge about injury risks and preventive measures, both at the community level and at the workplace, was found to be a common barrier to injury prevention, perceived to be mitigated by educational programmes. Together with previously published epidemiological evidence, the barriers and facilitators identified in this study offer useful basis to inform policy and practice.

## Strengths and limitations of this study

► Participants from diverse home environments, different work settings and different socioeconomic backgrounds yielded a breadth of views.
► This is the first study to have explored qualitatively the views and perceptions of the public about injury risks at home and at work in Nepal.
► The study is not able to provide perceptions about injury risks and preventive measures by injury type.

nearly 4.5 million people died from injuries in 2017, with a rate of disability-adjusted life years of 3267 per 100 000.[2] Of injury-related deaths, 90% occured in low-income and middle-income countries (LMICs), and in Nepal there has been an increase in injury-related deaths from 6.3% to 9.2% between 1990 and 2017.[3] Globally, road traffic injuries, falls, burns, poisonings and suicides are the leading causes of unintentional and intentional injuries.[4] According to the International Labour Organization, more than 2.78 million deaths per year are estimated to be due to occupational injuries or workplace disease.[5] In Nepal, 200 workers die and 20 000 workers suffer from workplace injuries yearly.[6]

Recent evidence for injuries occurring at home[7] identified parental supervision and teaching children about injury risks were facilitators, while barriers to child injury prevention were identified as parents' lack of anticipation of injury risks and perceiving injuries as inevitable events.[8] Culturally acceptable prevention measures, appropriate supervision arrangements, separation of hazards and training children and parents about safety were suggested by a study of community perceptions in Makwanpur, Nepal.[9] Two community-based studies conducted in rural Nepal emphasised that unintentional child injuries were thought to be due to coincidence, bad luck, witchcraft or

## INTRODUCTION

Injuries are a global health problem, although they are predictable and largely preventable.[1] According to global estimate,

ill fate.[9] [10] Likewise, in Bangladesh, child drowning was believed to be a result of ill fate and was unpreventable.[11] Rarely were the environmental and infrastructural factors thought by parents to be the cause of child injury.[9]

Workplace injuries are becoming a public health concern in all LMICs. One qualitative study undertaken in Bangladesh found that poor people were at greater risk of injury, employers were reluctant to take responsibility for workers and subcontracting workers was observed to increase the risk of injury.[12] Despite the high level of awareness about the use of personal protective equipment (PPE) among Nepali workers, there was poor practice of using PPE.[13] A qualitative study conducted among Nepali migrant workers suggested that workplace injuries were due to lack of health and safety regulations, risk-taking behaviour of workers and perceived work pressure.[14]

Human behaviour, being a complex phenomenon, is determined by environmental factors (such as social support/barriers, ability to change one's own environment), behavioural factors (such as skills, practice and self-efficacy) and personal factors (such as knowledge and perception).[15] To understand health and safety behaviours, one needs to understand how people perceive injury risks and what are the factors that influence their behaviours.[13] Little is known about how the people of Nepal perceive and deal with home and workplace injuries or their risk factors. This study explored a range of views and perception about injuries in a variety of settings and identified barriers and facilitators to injury prevention.

## METHODS
### Study design
We adopted a qualitative research methodology using focus group discussions and key informant interviews.

### Study setting and participants
The study took place in Makwanpur District of Nepal (see figure 1), which has a mixed topography similar to other districts in the country. The three administrative areas ('palikas') were selected purposively: Hetauda submetropolitan city (urban area), Thaha municipality (suburban area) and Bakaiya municipality (rural area). To ensure diversity in location, occupation, housing type and key government personnel, and to achieve the information power necessary to answer the research question,[16] the research team prepared a prespecified sampling framework where the key groups they wished to include were identified and listed. Members of the research team consulted with the existing networks and local government officers to identify the knowledgeable and experienced individuals and groups of participants. In each study area,

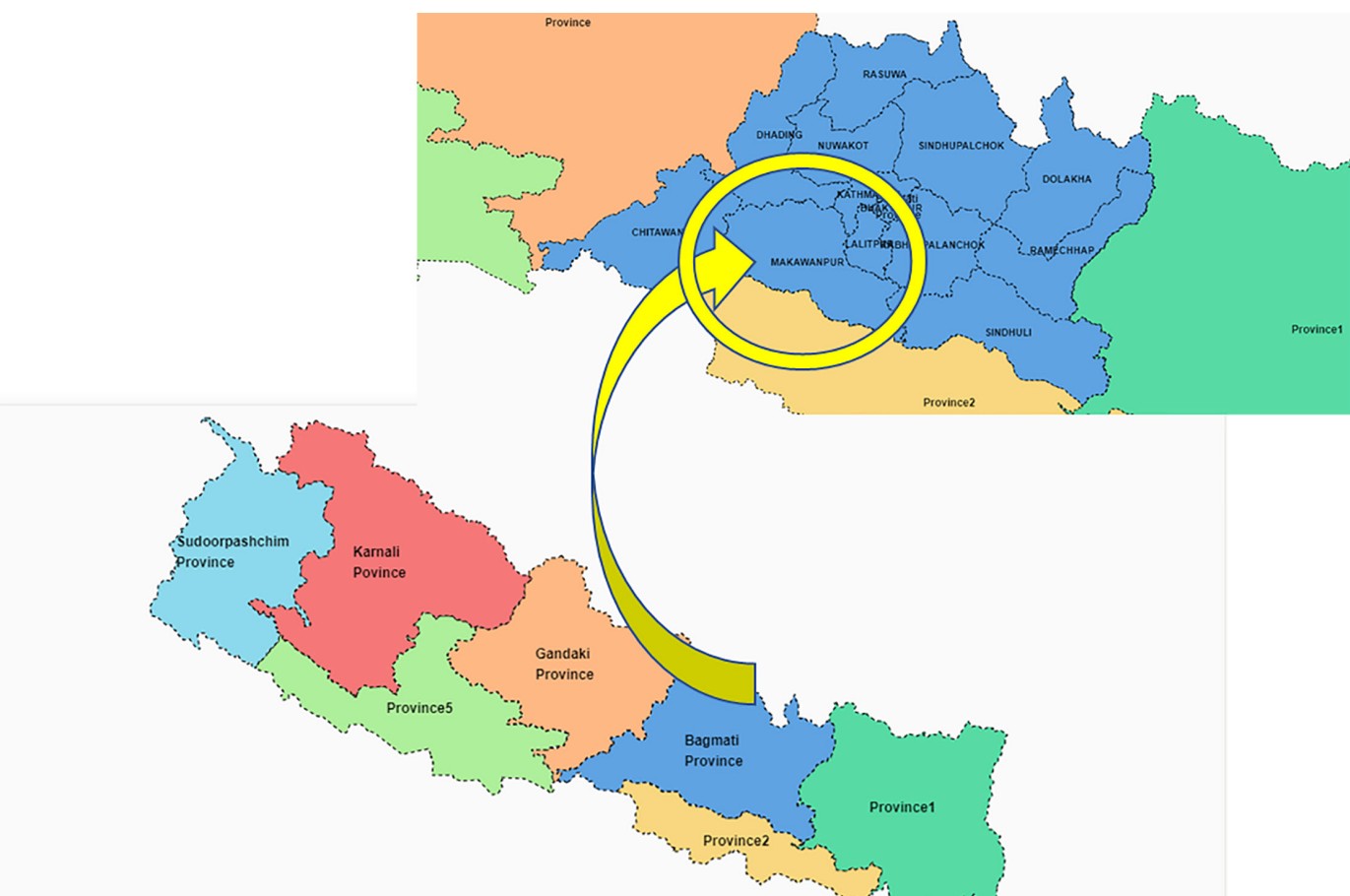

**Figure 1** Map of Nepal and Makwanpur District in Bagmati Province (source: https://nepalindata.com/).

a local non-governmental organisation (NGO), Mother and Infant Research Activities (MIRA), which has over 20 years of experience of working in Makwanpur District, Nepal, helped to identify prospective participants. The participants who met the inclusion criteria, were 18 years old or over and resident and/or working in one of the three palikas were approached by the research team. Identified individuals were invited to take part and were given a participant information sheet and consent form. Consent forms were completed by those who agreed to take part. The research team conducted all the focus groups and interviews at a location and time convenient to the participant groups and individuals in each study site. There was no adverse consequence for non-participation, and as a token of appreciation for their time and participation each participant was given 1000 Nepali rupees (equivalent to approximately £6).

### Data collection
The focus groups and interviews were conducted to explore the views and issues of people residing in varied accommodations and environments and workers in different workplaces. Interviews were conducted with key people, based on their role and nature of job.

A topic guide for the focus groups and a semistructured interview schedule were developed (see online supplemental files 1 and 2). These were developed based on previous work undertaken by the research team, pretested and finalised prior to data collection. All focus groups and interviews were conducted face-to-face by either SB or EJ. With consent, the focus groups and interviews were digitally audio-recorded and written notes were taken concurrently by a note taker to capture non-verbal communication. The interview schedule and topic guide asked questions about, for example, perception of injuries, risk factors, and barriers to and facilitators of injury prevention.

### Data management
Recordings were transcribed verbatim in Nepali and then translated to English with an aim to retain the original meaning of the statements. The transcriptions and translations were completed by experienced MIRA staff and verified by researchers (EJ and SB). Personal identifiable information from participants were removed from transcriptions and replaced with a unique identification code.

### Data analysis
Interview and focus group data were analysed using inductive thematic analysis to search for repeated patterns of meaning or themes in the data.[17] NVivo V.12 qualitative data analysis software was used to arrange the codes systematically and collate data relevant to each code.[18] Two transcripts were coded by both EJ and SB and discussed with a third researcher (TD) to agree an initial coding framework, which was then applied consistently across the transcripts. After coding all transcripts, the individual codes were reviewed and placed into clusters based on their similarities. The clusters were organised to develop candidate themes and barriers and facilitators were identified. Any discrepancies and differences in codes and themes developed were discussed with a third researcher (TD) and final themes were agreed.

### Patient and public involvement
We worked with a local NGO (MIRA) and local government officials to identify relevant groups and key personnel to recruit as participants in our study. No patients were involved in this study.

## RESULTS
### Participants' characteristics
We conducted nine focus groups and nine one-to-one interviews with a total of 83 participants across three municipalities. Out of ten participants that we approached, one refused to participate in the interview hence participation was 90% for the interviews. The participation for the focus groups was 100%. The average length of each focus group was 37 min and for the interview was 31 min. A total of six focus groups and six interviews were conducted in Hetauda submetropolitan city, two focus groups and two interviews in Bakaiya rural municipality, and one focus group and one interview in Thaha municipality. Participants in the focus groups were from similar backgrounds (table 1), whereas interviewees were representatives from different institutes and organisations (table 2).

### Themes
Six themes were identified from the focus group and interview data and these are described in the following sections. Relevant quotes are presented to illustrate the key themes.

#### Unsafe home and workplace environment
The physical infrastructure and home environment were perceived as common causes of unintentional injuries at home. The participants said that the lack of railings on stairs was the leading cause of fall injuries among people living in traditionally built rural houses, especially hazardous to children. Participants from urban areas raised concerns that elderly people sustained fall injuries due to slipping on marble or tiled bathroom floors. Burn injuries were common among children and women where firewood was used for cooking.

> While the kids are running in the balcony, they jump from there and get hurt. It's like that in these mud houses…there is no fence [railing]! (FGD 1, P8)

Regarding workplace injuries, different employees, like factory workers, healthcare staff and farmers, all highlighted that falls, cuts and burns were common injuries. Some described how they have to work in risky environments sometimes without safety equipment. One factory worker reported:

**Table 1** Characteristics of the focus group participants

| Focus group identification | Participants | Participants (n) | Age range of participants (years) | Gender |
|---|---|---|---|---|
| FG 1 | Residents in rural, traditionally built home | 8 | 21–55 | M: 4 F: 4 |
| FG 2 | Residents in rural, concrete/modern home | 9 | 21–55 | M: 5 F: 4 |
| FG 3 | Residents in urban area | 8 | 35–50 | M: 4 F: 4 |
| FG 4 | Haulage/truck drivers | 9 | 40–57 | M: 9 |
| FG 5 | Residents in slum area | 8 | 18–59 | M: 4 F: 4 |
| FG 6 | Skilled construction workers: plumber, welder, electrician | 8 | 21–50 | M: 8 |
| FG 7 | Farmers (agriculture, animal husbandry) | 8 | 37–51 | M: 4 F: 4 |
| FG 8 | Factory workers (cement factory) | 9 | 26–41 | M: 8 F: 1 |
| FG 9 | Trade professionals | 7 | 30–53 | M: 7 |

F, female; FG, focus group; M, male.

I have seen workers suffering from the pain due to heat. They have to take out rods from 300 degrees of heat by bare hands. They even don't wear gloves and boots. They use scissors [instead of tongs] … (KII 6)

Many participants explained that workers who are illiterate and suffer from poverty were those who worked under private contractors who are less likely to provide safety equipment.

### Inadequate supervision and monitoring

The participants highlighted the need for close supervision and monitoring of children and teenagers. One of the focus groups reported that there had been incidents of poisoning when children found liquids that looked like food or drink. Although parents said they were aware of this, they said that careful supervision was difficult

for working and single parents and therefore injuries occurred.

The parents have to work throughout the day to earn food and the children are guided either by the school or without anyone proper guidance. The parents are unknown about their children's behaviour. They simply grow without their parents' guidance… (KII 9)

### Perception that injuries are inevitable

With regard to both home and workplace injuries, participants highlighted that injuries occur by chance and unexpectedly and are normal in their situation. Some participants felt that one's luck is responsible for the occurrence of injuries. Acceptance of injuries as a normal part of life was a predominant finding; they

**Table 2** Characteristics of the interview participants

| Interviewee identification | Participants | Age range of participants (years) | Gender |
|---|---|---|---|
| KII 1 | President of mothers' group | 40–50 | F |
| KII 2 | Consultant in charge of hospital emergency department | 20–30 | F |
| KII 3 | Healthcare manager of municipality | 40–50 | M |
| KII 4 | Healthcare manager of submetropolitan city | 40–50 | F |
| KII 5 | Trade union representatives | 50–60 | M |
| KII 6 | Senior officer of Hetauda industrial area | 40–50 | M |
| KII 7 | Representative of transport workers | 40–50 | M |
| KII 8 | Rural municipality officer | 40–50 | M |
| KII 9 | Senior executive of education | 50–60 | M |

F, female; KII, key informant interview; M, male.

were uncertain about how injuries could be prevented. They believed that such injuries that occurred abruptly could not be prevented, despite them being careful when performing tasks.

> I feel that people get injured if they have misfortune or if the movement of planets that influences the destinies of people is not good or else, they [people] won't get injured. Even if we are careful, we might get injured due to our bad fortune. (FG 5, P2)

> It's like this. In the workshops the cuts are normal. We need to play with the metals. While doing that [playing with metals] the sharp edges of it [metals] can give cuts to us. We also get normal cuts when we are cutting down the pipes. (FGD 6_P2)

### Safety takes low priority: financial and behavioural considerations

Most participants said that when they were at home they did not think about safety. Carelessness when completing tasks in a hurry was the most common reason for injuries occurring at home and workplace.

> Most of the injuries take place due to our carelessness. Like we throw nails and other sharp materials here and there…. Another reason is … performing any kind of activities hurriedly while walking … (FG 2, P2)

Participants from rural villages confessed that financial constraints hindered them from implementing safety measures like installing railings to balconies as they struggled to meet their families' basic needs.

> Due to this economic barrier, we are unable to manage everything… (FG 1, P5)

Participants described the common practice of employers who pressurised their workers to work fast without providing PPE. Inadequate capital was one reason for employers for not providing PPE or installing safer, automatic machines.

> …the manufacturers of construction factories like grill factories are more concerned about how to increase their production leading the workers to work fast and carelessness at work. (KII 5)

The workers stated that not only did they have to work without PPE, especially the contracted workers, but most of them worked carelessly, misusing or not using safety equipment even when the PPE was available. Overconfidence and feeling that nothing would happen to them led the workers to take such risks.

> Even if we have safety equipment, we get careless and do not use them thinking that the work we are about to perform just takes 2 minutes of time and we won't require it [safety equipment]. But, in that 2 minutes of time any incident may occur. (FG 7, P3)

Senior personnel emphasised that it was the role of supervisors to consider safety, raise awareness and orient the workers to the dangerous aspects of their work.

> What I would say is once you recruit the person, he/she should be taught everything. Every industry should manage this and conduct orientation. The industry should make employees learn about the culture of work. But this culture isn't practised. (KII 6)

### Safety education and training

Additional to the behavioural and financial aspects, participants reported that lack of awareness was a major contributing factor that led to home and workplace injuries; illiteracy and lack of understanding about injuries were prevalent. Almost all participants voiced the need for an awareness programme, that it would be a crucial enabling factor for injury prevention, both at home and at the workplace.

> In case of home, there should be awareness program because they [community people] don't know. They should be educated such that they can bring change in their behaviour. It is because they don't understand. (KII 3)

Regarding home injury, many participants stated that a programme to reduce injuries at home should be conducted by local community groups and their leaders, such as mothers' groups. Participants believed that the effectiveness of programmes relied on being led by injury experts in collaboration with NGOs, international non-governmental organisations (INGOs) and local government.

> If the outsiders go and tell them [community people] they might listen. But if we locals tell them they won't listen to us. (FG 2, P4)

Participants also felt that such injury prevention programmes should be delivered to school children, teachers, parents and community leaders using social media or community mobilisation.

> For that we must gather all the family members in a place and discuss about this matter [injuries]. We must discuss about how injuries take place and how we can reduce it [injuries]. (FG 3, P7)

Factory workers raised concerns about the lack of institute to train workers, to make them aware of occupational health and safety (OHS). Skilled workers and labourers highlighted the need for safety training, that it could be provided and monitored by industries, NGOs and the Ministry of Labour. There was a consensus among factory workers that training needed to be inclusive of all workers, company and contract staff, including the employers, and that it should be done regularly, addressing any technological updates and human tendency to forget. Pamphlets or brochures with brief messages would raise awareness among workers.

The main important thing is to [make aware] the employer, those who employ the workers. Basically, they should be aware of the health of the workers. We should also be able to make the workers understand that, if you are healthy then you can do lot of work… (KII 5)

### Government-led safety programmes and enforcement

Almost all participants thought that the government's role was pivotal in injury prevention activities. They thought that the government lacked focus on injury prevention activities due to lack of injury data. Most participants believed that the local government authorities needed to take a lead on injury prevention activities. On probing what would work, they thought that collaboration with NGOs, INGOs and injury experts, strategic planning could be achieved. Participants provided examples of how the municipality was working in partnership with some organisations for community health development and women empowerment. For example, the municipality was working with research organisations to prioritise injury prevention programmes.

The local government along with the NGO/INGOs have mobilized the mother's group… The local government should launch such a program [injury prevention] which will enhance them [mothers] and help to promote the injury related activities. (KII 9)

The participants highlighted the role of the central government could be to formulate uniform policies, with strict enforcement of rules and regulations to prevent injuries both at home and at the workplace. Periodic monitoring visits to industries undertaken by the government and private sectors were felt to be mandatory to ensure safety standards were being followed.

The government of Nepal should formulate standard protocol to prevent injuries at industries. If there isn't environment as mentioned in the protocol then they shouldn't be renewed and the industry should be closed immediately. (KII 3)

### Barriers to and facilitators of injury prevention

The key barrier to injury prevention identified by participants was the lack of knowledge about injury risk and preventive measures, both at the community level and at the workplace. Other barriers included safety not being a priority and the inability to improve the safety of the home or working environments due to financial constraints. The barriers, particularly for child injury prevention, included the inability of parents to provide adequate supervision. Participants identified that safety at home and at the workplace could be improved at a community level through educational programmes to raise awareness about how injuries could be prevented. At the workplace,

health and safety training could be provided to employers and employees and safety laws introduced with rigorous enforcement. The involvement of external agencies and of government authorities could facilitate the implementation and effectiveness of such programmes, training and legislation.

## DISCUSSION

This study found that injuries were perceived as a problematic issue and preventive measures have been neglected across different settings.

### Home injuries

In our study, the home environment and socioeconomic aspects were highlighted as the main contributing factors for injuries such as falls, burns and poisonings. Unsafe buildings and cooking on open fires have been identified previously as injury hazards,[19] and in a countrywide population-based study conducted in Nepal, falls were found to be the leading cause of injury.[20] A household survey that assessed home hazards for child injury in rural Nepal found that 98% of households did not have protective railings on stairs, more than 80% of households had no window bars, and 50% of households lacked a protective barrier on their balconies.[21] Similar to the findings from Bangladesh, the study participants perceived financial constraints to be one of the barriers to prevention of home injuries.[22]

The findings from our study support those from previous epidemiological evidence where children were found to be at high risk of burns and that the home was hazardous.[23 24] Participants in our study provided the context and reported reasons for some of the more common injuries identified by other studies. For example, open fires for cooking was unsafe[25] and the majority of households (61%) in rural areas of Nepal had chemicals or fertilisers within the reach of children.[21]

Some parents believed that injuries were a normal part of child development, bad luck, witchcraft or ill fate. This fatalism, in relation to injury, has been reported in previous studies conducted in LMICs,[7 26 27] but no method to address this belief has been developed. A qualitative study in Nepal showed that when people are unable to rationalise the cause then the concept of fate is ascribed.[28] This lack of knowledge in the community has been reported in previous studies conducted in Nepal.[9 10] In line with previous studies, the participants of this study also highlighted the need for interventions at all levels, such as government, NGOs and the local community.[22]

### Workplace injuries

Minor injuries were reported as being common occurrences within the workplace, but fatal injuries and those that led to long-term disability were also reported. Study participants believed that poverty and illiteracy were interlinked and were the root cause of workplace injuries. This might be because employers tended to hire the poorer

workers who are ready to take on risky jobs but with a relatively higher wage[12] or that, similar to our study, carelessness was a significant causal factor, as found in one study of Iranian workers.[29]

Study participants stressed the need to raise awareness among workers about the importance of PPE use via regular and periodic OHS educational programmes. A study in India demonstrated that the understanding of occupational hazards was significantly associated with the literacy status of workers, and delivering occupational health educational programmes augmented the understanding of OHS.[30] Another study suggested that such programmes should be delivered together with behaviour change counselling,[31] but our study participants believed that awareness programmes, with adequate PPE and enforcement of occupational standards, would suffice for adopting safe workplace practices. One study found that high levels of awareness of occupational hazards and use of PPE did not lead to the actual use of PPE.[32] Further exploration is needed about why workers do not use PPE despite the availability and training. Unlike other studies which highlighted lack of user-friendly PPE and ineffective PPE,[33 34] our study found both an absence of formal training prior to commencing work and lack of PPE which created environments conducive to injuries.

In Nepal, the 1992 Occupational Health and Safety law did not specify how to evaluate or enforce the law.[35] After the revision in 2017, it explicitly described the roles of employees and employers in adhering to OHS standards, yet legislation is silent on the supervision and monitoring of the implementation status.[36] In line with other studies, study participants raised concerns about poor enforcement of safety rules and regulations by government agencies.[37] While the participants felt that this was needed, they also felt that neither party prioritised OHS; this was felt to be due to the lack of inspection and enforcing penalty or fines for infringing the rules. This study highlights the need for a coordinated approach to injury prevention. Through the unified efforts of the government, NGOs, communities and other professionals, social inequalities, enforcement issues and educational activities could be addressed and delivered.

### Strengths and limitations

To the best of our knowledge, this is the first qualitative study that reported information about home and workplace injuries among people living in varied settings and involved in various occupations in Makwanpur District in Nepal. The study design ensured a diversity of perspectives about both home and workplace injuries from people living in different home environments, those working in different occupations, from different levels of authority and from different socioeconomic backgrounds. A limitation of the study is that it does not provide information on injury-specific interventions, but it does present a sound basis from which injury researchers can explore specific injury risks and identify measures to remove them.

### Future directions and possible solutions

Based on participants' suggestions, home-based visits and awareness campaigns could be potential interventions for home injury prevention. Regarding prevention of workplace injuries, our study found that there was a need for regular workplace training for both employees and employers on safety measures, along with a mechanism to ensure levels of knowledge were maintained. Our study also stressed the need for a cultural shift so that greater self-efficacy to keep oneself and one's family and work colleagues safe becomes the norm. Findings from our study, collated with those of epidemiological studies, will be a good starting point to inform policy-level discussions. This would enable formulation and enforcement of policies and strategies related to home and workplace injury prevention.

### CONCLUSION

The findings of this study highlighted that both home and workplace injuries are complex and multifactorial and are influenced by personal, situational and environmental factors. Most importantly, lack of knowledge about injury risks and preventive measures, both at the community level and at the workplace, was found to be a common barrier to injury prevention. The introduction and implementation of educational home safety programmes delivered within the community and as an occupational safety programme within the workplace would be welcomed. Together with previously published epidemiological evidence, the perceptions of risk and perceived barriers to and facilitators of safety identified in this study provide a useful basis on which policy makers can establish their decision-making when addressing home and workplace injuries in Nepal.

**Acknowledgements** We acknowledge the support of all three palika authorities in approving the conduct of this study. We are grateful to all the staff at Mother and Infant Research Activities who lent their 20 years of research field experience to our study, supporting recruitment of the study participants and providing the logistics to conduct data collection. We are thankful to all the participants of our study who consented to participate and share their experience. We would like to acknowledge the support of the wider research team in the NIHR Global Health Research Group at the Nepal Injury Research UK (especially Emer Brangan) and the Nepal team and administrative support of Kathmandu Medical College who have supported the study.

**Contributors** SB, TD, JM and SKJ contributed to the conception and design of the study. SB drafted the protocol design, methods and data analysis plan. SB and EJ contributed to data collection with support from SRM and DA. SB and EJ led the analysis, interpretation of data and drafted the manuscript. EJ and SB drafted and finalised the manuscript with equal contributions. All authors contributed to drafts and approved the final manuscript.

**Funding** This research was funded by the National Institute for Health Research (NIHR) (ref: 16/137/49) using UK Aid from the UK Government to support global health research. The views expressed are those of the author(s) and not necessarily those of the NIHR or the UK Department of Health and Social Care.

**Map disclaimer** The depiction of boundaries on the map(s) in this article does not imply the expression of any opinion whatsoever on the part of BMJ (or any member of its group) concerning the legal status of any country, territory, jurisdiction or area or of its authorities. The map(s) are provided without any warranty of any kind, either express or implied.

**Competing interests** None declared.

**Patient consent for publication** Not required.

**Ethics approval** Ethical approval was obtained from the Nepal Health Research Council (reg. no. 779/2018) and the University of the West of England, Bristol, Faculty Research Ethics Committee (reference: HAS.19.04.157). Institutional approval from each municipality office was also obtained for conducting the study in that location.

**Provenance and peer review** Not commissioned; externally peer reviewed.

**Data availability statement** Data are available upon reasonable request. Data sets are de-identified transcripts of nine interviews and nine focus groups (in Nepali translated to English).

**Author note** EJ and SB are joint first authors

**ORCID iD**
Elisha Joshi http://orcid.org/0000-0002-3534-276X

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
