## [Reviewer comments · BMJ Open]

ARTICLE DETAILS

TITLE (PROVISIONAL)	Perceptions of injury risk in the home and workplace in Nepal: a qualitative study
AUTHORS	Joshi, Elisha; Bhatta, Santosh; Deave, Toity; Mytton, Julie; Adhikari, Dhruva; Manandhar, Sunil Raja; Joshi, Sunil Kumar

VERSION 1 – REVIEW

REVIEWER	Janneke Berecki-Gisolf Monash University
REVIEW RETURNED	23-Oct-2020

GENERAL COMMENTS	In their study titled 'Barriers and enablers for injury prevention at home and workplace in Nepal: a qualitative study', the authors provide an overview of a series of interviews and focus groups conducted in three areas in Nepal. The aim of the study was to explore attitudes and perceived barriers to home and workplace injury prevention in Nepal. The authors have clearly collected a wealth of data in this study, and the study introduction as well as the results make a strong case for the need for improved injury prevention. There are, however, some concerns with the manuscript in its current form. The main study limitation is the lack of focus. 'Little is known about how people of Nepal perceive and deal with home and workplace injuries, its risk factors and preventive measures during their daily life' is the study justification, i.e. gap in knowledge. 'This study explored a range of views on home and workplace injuries and identified barriers and facilitators to injury prevention' is the study purpose. This research would have benefited from a more targeted, focused aim. Lacking this, the results are wide-ranging and expansive. In particular, combining in one study: - Workplace vs. home safety- Adult vs. child injury- Rural vs. regional injury- Unintentional vs. intentional (assault, self harm) injury is too much to report on in one paper. At the very least, I would suggest removing reference to intentional injury from this paper (it is quite distinct from unintentional injury and requires a different approach to prevention) and splitting the results into home vs. workplace. Child injuries are likely to be more relevant to the home setting. Much of the required information for a qualitative study is missing from the methods section. How representative are the selected regions, of the Nepal population overall? What are 'municipality officers'? How were participants selected and approached? Was there any incentive (monetary or otherwise) or alternatively, adverse consequences for non-participation? Was the participation rate 100% (except for the participant who refused an
---

	interview)? From the results, I can see that the focus groups were workers and residents, whereas the interviews were conducted among persons in official roles with relevance to injury prevention. Please describe this approach in the methods section. Please also justify the number of interviews and focus groups – was this continued until saturation was reached (i.e. the same themes kept emerging and further interviewing no longer provided additional insights)? In the results section, the examples don't always fit within the identified themes. 'Unsafe home and workplace environment' has an example of intentional injury resulting from family violence. This is very different to the workplace example (heat exposure in the workplace) given under the same header. Splitting the results into home vs. work and removing reference to intentional injuries will help retain focus in this study. In the theme 'Safety takes low priority...', one of the examples refers to 'destiny'. Indeed, the sense that injuries are not preventable; 'fate'; 'destiny'; seems to come up several times and this, I think, is another theme that should be listed separately. Other examples in 'Safety takes low priority...' refer to an absence of safety culture in the workplace. A final comment refers to the title and discussion: It seems that 'facilitators' and 'enablers' are not captured but rather, suggestions for future plans or potential solutions to the injury problem in Nepal. Based on the study purpose and title, it would be expected that perceived 'facilitators' and 'enablers' refer to the current situation, just as 'barriers' refer to the current situation. Future directions and potential solutions should be discussed separately to this.
--	---

REVIEWER	Amy Peden School of Population Health, UNSW Sydney
REVIEW RETURNED	11-Nov-2020

GENERAL COMMENTS	Thank you for the opportunity to review manuscript id bmjopen-2020-044273 entitled "Barriers and enablers for injury prevention at home and workplace in Nepal: a qualitative study" which was submitted for consideration for publication in BMJ open. This study is well-written and adds to the limited published data on injury prevention in Nepal. I commend the authors for undertaking the study, and in particular, the use of a qualitative method. I have very few suggestions for improvement. Please see below for some comments relating to specific sections of the manuscript that I feel need to be addressed in a very minor revision of this manuscript. Comments Abstract Design: line 11 – were interviews one on one? I am assuming so but it would be good to specify here. Or state the (9) participants after interviews in the participants sections Main manuscript Introduction Line 6 – you may like to use the newly released GBD estimates of injury (https://injuryprevention.bmj.com/content/26/Supp_1/i96.long) and add in the morbidity burden also Line 20 – perhaps have been identified as facilitators rather than were considered facilitators? This sentence reads a little strangely to me Methods
--

	Page 5 – lines 23-26 – suggest adding a map here as a figure, for people not familiar with Nepal to understand the geography of the country and the location of the study sites. Page 5 – Line 41- are you able to upload the interview guide as a supplementary file please? Were participants reimbursed for participation. Can you add some details on recruitment and reimbursement to the methods? Results Page 7 line 60 & page 8 line 3 – this sentence about the recent change in social structure and men working outside of Nepal is not a complete sentence the way it is written. Suggest change that the recent change to with the recent change Page 9, line 40 – you have already explained the PPE acronym earlier in the manuscript so you can just use PPE here Page 10 – line 3 – not sure what is meant by “word supervisors” should word be work? Please check and revise
--	---

VERSION 1 – AUTHOR RESPONSE

Reviewer comments	How we have addressed	Page no.
Reviewer 1		
In their study titled ‘Barriers and enablers for injury prevention at home and workplace in Nepal: a qualitative study’, the authors provide an overview of a series of interviews and focus groups conducted in three areas in Nepal. The aim of the study was to explore attitudes and perceived barriers to home and workplace injury prevention in Nepal. The authors have clearly collected a wealth of data in this study, and the study introduction as well as the results make a strong case for the need for improved injury prevention. There are, however, some concerns with the manuscript in its current form. The main study limitation is the lack of focus. ‘Little is known about how people of Nepal perceive and deal with home and workplace injuries, its risk factors and preventive measures during their daily life’ is the study justification, i.e. gap in knowledge. ‘This study explored a range of views on home and workplace injuries and identified barriers and facilitators to injury prevention’ is the study purpose. This research would have benefited from a more targeted, focused aim. Lacking this, the results are wide-ranging and expansive. In particular, combining in one study:  - Workplace vs. home safety -  Adult vs. child injury - Rural vs. regional injury - Unintentional vs. intentional (assault, self-harm) injury is too much to report on in one paper. At the very least, I would suggest removing reference to intentional injury from this paper (it is quite distinct from unintentional injury and requires a different approach to prevention) and splitting the	Thank you so much for your insightful suggestions. We have amended and simplified our paper to reflect your feedback that the paper lacks focus. We have done the following:  • Removed ‘barriers and enablers’ from the title and renamed it, ‘Perceptions of injury risk in the home and workplace in Nepal: a qualitative study.’ • Reworded the aim to clarify the purpose • Removed all references to intentional injury • Added a section in the methods to be explicit about the 	1-15

results into home vs. workplace. Child injuries are likely to be more relevant to the home setting.	identification of barriers and facilitators from the themes derived from the data. The analysis was planned to look at perception of risk across settings  • Removed 'column 1' and the 'location' column from tables 1 & 2 to avoid confusion regarding the purpose of the study: to explore perceptions of injury and not to compare the home with the workplace or to compare urban with rural settings • Added a sub-title in the discussion section. 'Future directions and possible solutions' to 'signpost' the paragraph. 	
Much of the required information for a qualitative study is missing from the methods section. How representative are the selected regions, of the Nepal population overall?	Thank you for your valuable suggestion. In response, we have added more information about the population and selected regions in the 'Study Setting and Participants' section of the method section (page 4).	4-5
What are 'municipality officers'?	Thank you for highlighting this potentially confusing term. We have renamed municipality officers as local government officers.	4
How were participants selected and approached? Was there any incentive (monetary or otherwise) or alternatively, adverse consequences for non-participation?	Thank you for your questions. We have included additional details	4

	in the Methods section (page 4) of the paper, as below: 'To ensure diversity of location, occupation, housing type and key government personnel, and to achieve the information power necessary to answer the research question,(1) the research team prepared a pre-specified sampling framework where we identified and listed key groups we wished to participate. Members of the research team consulted with the existing networks and local government officers to identify the knowledgeable and experienced individuals and groups of participants. In each study area, a local non- governmental organisation (NGO), Mother and Infant Research Activities (MIRA) who have over 20 years of experience of working in Makwanpur district, Nepal, helped to identify prospective participants. The participants, who met the inclusion criteria were 18 years old or over and resident and/or working in one of the three palikas were approached by the research team. Identified individuals were invited to take part, were given a participant information sheet and consent form. Consent forms were completed for those who agreed to take part. The research team conducted all the focus groups and interviews at a location and	
--	---	--

	time convenient to the participant groups and individuals in each study site. There was no adverse consequence for non-participation and as a token of appreciation for their time and participation, each participant was given 1000 Nepali Rupees (equivalent to approximately 6 British Pounds).'	
Was the participation rate 100% (except for the participant who refused an interview)? From the results, I can see that the focus groups were workers and residents, whereas the interviews were conducted among persons in official roles with relevance to injury prevention. Please describe this approach in the methods section	Thank you for your question and recommendation. The participation was 100% (mentioned in the results section). We have added the text below into the paper regarding the participants for focus groups and interviews: 'The focus groups (FGs) and interviews were conducted to explore views and issues of people residing in varied accommodation and, environments and different workers in their workplaces. Interviews were conducted with key people based, on their role and, nature of their job.'	6
Please also justify the number of interviews and focus groups – was this continued until saturation was reached (i.e. the same themes kept emerging and further interviewing no longer provided additional insights)?	Thank you for your comment. The number of interviews and focus groups were determined by the need to ensure we engaged a diverse range of households and professions across the participating palikas. As described in the new text in the methods (and replicated in the answer to the previous question), we constructed a sampling framework to guide our	5

	recruitment of participants. We did not use the concept of saturation to determine the number of participants but instead used the concept of information power as described by Malterud et al. (2016) and referenced in the newly inserted text.	
In the results section, the examples don't always fit within the identified themes. 'Unsafe home and workplace environment' has an example of intentional injury resulting from family violence. This is very different to the workplace example (heat exposure in the workplace) given under the same header. Splitting the results into home vs. work and removing reference to intentional injuries will help retain focus in this study.	Thank you for your comment. We have removed references to intentional injury description in this paper and re-structured the results to add under the theme: 'Unsafe home and workplace environment'. The study was planned to look at the perceptions of injury risk across settings and to identify particular issues for any particular settings. We did not set out to compare home vs workplace, Therefore, we have amended the text to clarify this.	7-8
In the theme 'Safety takes low priority...', one of the examples refers to 'destiny'. Indeed, the sense that injuries are not preventable; 'fate'; 'destiny'; seems to come up several times and this, I think, is another theme that should be listed separately. Other examples in 'Safety takes low priority...' refer to an absence of safety culture in the workplace.	Thank you for your suggestion. As per your advice, we have added another theme "Injuries are inevitable" and separated it from theme "Safety takes low priority..."	8-9
A final comment refers to the title and discussion: It seems that 'facilitators' and 'enablers' are not captured but rather, suggestions for future plans or potential solutions to the injury problem in Nepal. Based on the study purpose and title, it would be expected that perceived 'facilitators' and 'enablers' refer to the current situation, just as 'barriers' refer to the current situation. Future directions and potential solutions should be discussed separately to this.	Thank you for your suggestion. The barriers and facilitators illustrated on page 7-11 have been identified from the data; these are current ones. The methods section, as described above, has had a sentence added and a few amendments have been made to ensure this is clear.	6, 7-11 and 14

	As per your advice, we have added a sub-heading, 'Future directions and possible solutions' in the discussion section to highlight that paragraph.	
Reviewer 2		
Abstract Design: line 11 – were interviews one on one? I am assuming so but it would be good to specify here. Or state the (9) participants after interviews in the participants sections	Thank you for very much for your comment. All the interviews conducted were one to one. We have added this into the Abstract.	2
Main manuscript Introduction Line 6 – you may like to use the newly released GBD estimates of injury (https://injuryprevention.bmj.com/content/26/Supp_1/i96.long) and add in the morbidity burden also	Thank you for recommendation. We have included the newly released GBD estimate of injury and morbidity burden of global and Nepal in the introduction section (page 3): 'According to the global report 2017, nearly 4.5 million people die from injuries and disability-adjusted life years (DALYs) is 3267 per 100,000.(2)'	3
Line 20 – perhaps have been identified as facilitators rather than were considered facilitators? This sentence reads a little strangely to me	Thank you for highlighting that error. We have changed that to 'identified as facilitators.'	3
Methods Page 5 – lines 23-26 – suggest adding a map here as a figure, for people not familiar with Nepal to understand the geography of the country and the location of the study sites.	Thank you very much for this suggestion. A map of Nepal, with Makwanpur, the study site, has been identified and added to the Methods section.	5
Page 5 – Line 41- are you able to upload the interview guide as a supplementary file please? Were participants reimbursed for participation. Can you add some details on recruitment and reimbursement to the methods?	Thank you for your suggestion. The topic guide and semi-structured interview schedule used in this study have been uploaded as supplementary files. Details of reimbursement and recruitment have been	4-5

	added in the Methods section as: 'The participants were given 1000 Nepali Rupees (equivalent to approximately 6 British Pounds) as a token of appreciation for their participation and time in the study.'	
Results Page 7 line 60 & page 8 line 3 – this sentence about the recent change in social structure and men working outside of Nepal is not a complete sentence the way it is written. Suggest change that the recent change to with the recent change	Thank you for so much for your comment. In response to reviewer 1's recommendation to remove reference to intentional injuries, this section has been removed.	NA
Page 9, line 40 – you have already explained the PPE acronym earlier in the manuscript so you can just use PPE here	Thank you for the suggestion. We have used the acronym PPE in the remaining manuscript.	8-12
Page 10 – line 3 – not sure what is meant by “word supervisors” should word be work? Please check and revise	Thank you for this helpful observation. We have changed the typographical error from 'word' to 'work'.	9
Editor		
Required amendments will be listed here; please include these changes in your revised version: - We have implemented an additional requirement to all articles to include 'Patient and Public Involvement' statement within the main text and your main document. Please refer below for more information regarding this new instruction: Patient and Public Involvement: Authors must include a statement in the methods section of the manuscript under the sub-heading 'Patient and Public Involvement'. This should provide a brief response to the following questions: How was the development of the research question and outcome measures informed by patients' priorities, experience, and preferences? How did you involve patients in the design of this study? Were patients involved in the recruitment to and conduct of the study? How will the results be disseminated to study participants?	Thank you for your advice. We have added a section on patient and public involvement as requested. "Patient and Public Involvement" We worked with a local NGO (MIRA) and local government officials to identify relevant groups and key personnel to recruit as participants in our study. No patients were involved in this study.	6

For randomised controlled trials, was the burden of the intervention assessed by patients themselves? Patient advisers should also be thanked in the contributorship statement/acknowledgements. If there is no patient involved in the study, please state "No patient involved" under the sub-heading 'Patient and public involvement'.		
---	--	--

References

1. Malterud K, Siersma VD, Guassora AD. Sample size in qualitative interview studies: guided by information power. *Qual Health Res.* 2016;26(13):1753-60. DOI: <https://doi.org/10.1177%2F1049732315617444>.
2. James SL, Castle CD, Dingels ZV, et al. Global injury morbidity and mortality from 1990 to 2017: results from the Global Burden of Disease Study 2017. *Inj Prev.* 2020. DOI: <http://dx.doi.org/10.1136/injuryprev-2019-043494>.

VERSION 2 – REVIEW

REVIEWER	Janneke Berecki Monash University, Australia
REVIEW RETURNED	09-Feb-2021

GENERAL COMMENTS	The authors have addressed my previous comments very well. I have two remaining minor comments: Results, page 7, lines 10-11: One participant refused to participate in the interview, participation rate was 100%. This seems a contradictory statement; please clarify. Results, page 9, line 9. Heading: Injuries are inevitable. Perhaps add 'Perception that..' to this heading, or 'Acceptance of injuries as a normal part of life', as stated in the corresponding paragraph.
--

REVIEWER	Amy Peden UNSW Sydney, Australia
REVIEW RETURNED	01-Feb-2021

GENERAL COMMENTS	Thank you for the opportunity to review this revision. I think you have done a great job and addressed all my concerns. Suggest a thorough proof read to pick up any last typographical or grammatical errors but I am comfortable in accepting the paper for publication in its current form.
---

VERSION 2 – AUTHOR RESPONSE

Response to Reviewer

S.No.	Reviewer Comments	Response
Reviewer 1		
1.	Results, page 7, lines 10-11: One participant refused to participate in the interview, participation rate was 100%. This seems a contradictory statement; please	Thank you so much for highlighting the issue. We have clarified it as:

	clarify.	“Out of ten participants that we approached, one participant refused to participate in the interview; participation was 90% for interview. The participation for the focus groups was 100%.”
2.	Results, page 9, line 9. Heading: Injuries are inevitable. Perhaps add 'Perception that..' to this heading, or 'Acceptance of injuries as a normal part of life', as stated in the corresponding paragraph.	Thank you so much for your suggestion. We agree with you and revised the theme title as: “Perception that injuries are inevitable”.
Reviewer 2		
1.	Thank you for the opportunity to review this revision. I think you have done a great job and addressed all my concerns. Suggest a thorough proof read to pick up any last typographical or grammatical errors but I am comfortable in accepting the paper for publication in its current form.	Thank you so much for your suggestion. We have done a proof read to remove grammatical errors.